# Augmented Reality and Virtual Reality in Dentistry: Highlights from the Current Research

Sidra Fahim [1], Afsheen Maqsood [2], Gotam Das [3], Naseer Ahmed [4,*], Shahabe Saquib [5], Abhishek Lal [4], Abdul Ahad Ghaffar Khan [6] and Mohammad Khursheed Alam [7,8,9,*]

1   Department of Oral Medicine, Altamash Institute of Dental Medicine, Karachi 75500, Pakistan;
    sidra.raf@gmail.com
2   Department of Oral Pathology, Bahria University Dental College, Karachi 07557, Pakistan;
    afsheenmaqsood.bumdc@bahria.pk
3   Department of Prosthodontics, College of Dentistry, King Khalid University, Abha 61341, Saudi Arabia;
    gmenghwar@kku.edu.sa
4   Department of Prosthodontics, Altamash Institute of Dental Medicine, Karachi 75500, Pakistan;
    abhishekdarshan@yahoo.com
5   Department of Periodontics and Community Dental Sciences, College of Dentistry, King Khalid University,
    Abha 61341, Saudi Arabia; sshahabe@kku.edu.sa
6   Department of Oral and Maxillofacial Surgery, College of Dentistry, King Khalid University,
    Abha 61341, Saudi Arabia; abahkhan@kku.edu.sa
7   Department of Preventive Dentistry, College of Dentistry, Jouf University, Sakaka 72345, Saudi Arabia
8   Center for Transdisciplinary Research (CFTR), Saveetha Dental College, Institute of Medical and Technical Sciences,
    Saveetha University, Chennai 600077, India
9   Department of Public Health, Faculty of Allied Health Sciences, Daffodil International University,
    Dhaka 1341, Bangladesh
*   Correspondence: drnaseerahmed@altamash.pk (N.A.); mkalam@ju.edu.sa (M.K.A.)

**Abstract:** Many modern advancements have taken place in dentistry that have exponentially impacted the progress and practice of dentistry. Augmented reality (AR) and virtual reality (VR) are becoming the trend in the practice of modern dentistry because of their impact on changing the patient's experience. The use of AR and VR has been beneficial in different fields of science, but their use in dentistry is yet to be thoroughly explored, and conventional ways of dentistry are still practiced at large. Over the past few years, dental treatment has been significantly reshaped by technological advancements. In dentistry, the use of AR and VR systems has not become widespread, but their different uses should be explored. Therefore, the aim of this review was to provide an update on the contemporary knowledge, to report on the ongoing progress of AR and VR in various fields of dental medicine and education, and to identify the further research required to achieve their translation into clinical practice. A literature search was performed in PubMed, Scopus, Web of Science, and Google Scholar for articles in peer-reviewed English-language journals published in the last 10 years up to 31 March 2021, with the help of specific keywords related to AR and VR in various dental fields. Of the total of 101 articles found in the literature search, 68 abstracts were considered suitable and further evaluated, and consequently, 33 full-texts were identified. Finally, a total of 13 full-texts were excluded from further analysis, resulting in 20 articles for final inclusion. The overall number of studies included in this review was low; thus, at this point in time, scientifically-proven recommendations could not be stated. AR and VR have been found to be beneficial tools for clinical practice and for enhancing the learning experiences of students during their pre-clinical education and training sessions. Clinicians can use VR technology to show their patients the expected outcomes before the undergo dental procedures. Additionally, AR and VR can be implemented to overcome dental phobia, which is commonly experienced by pediatric patients. Future studies should focus on forming technological standards with high-quality data and developing scientifically-proven AR/VR gadgets for dental practice.

**Keywords:** augmented reality; virtual reality; dentistry; education; health

## 1. Introduction

The field of dentistry has recently experienced many technological advancements that have changed the dimensions of many subspecialties of dentistry. To further improve and enhance dental education and the clinical application of dentistry, new innovations have been introduced. These include augmented reality (AR) and virtual reality (VR), which have been introduced and studied with the goal of improving dentistry. The word "virtual" in the field of computing and technology refers to something that appears to exist without being physically present in the real world. In dentistry, software is used to simulate dentofacial structures in real-time. The functionality of these anatomical structures is also simulated for a sophisticated yet complete virtual experience.

The commonly-used traditional digital technologies operate by using a digital scanner to obtain a digital image, then the operator uses the digital image to modify the different dental aspects of the patients, and then finally, the modifications are transferred to the digital wax-up. Virtual reality, however, takes laser scans of the patient's teeth and other oral and extraoral structures as required and feeds them into a computer that forms a 3D model of the same, which is then loaded into a simulator. Dental surgeons/dental students using this simulator can practice and do evaluations before the procedure is performed on a real patient in various specialties of dentistry, i.e., oral and maxillofacial surgery, orthodontics, implantology, restorative dentistry, dental public health, and dental education [1,2]. Automated data recording is integrated into this system, making post-operative analysis and self-assessment possible for the users.

Augmented reality mainly aims to improve the clinical practice in the field of dentistry as the clinical information that is generated can be directly visualized on the patient, combining the real world with the digital world. The primary use of augmented reality in dentistry comprises the use of digital information to improve reality, which allows effective communication between the patients and dentists through the use of videos, pictures, and three-dimensional models.

AR is also an interactive technology but is distinct from virtual reality in the way that the user interacts with an integral image of the patient's teeth/anatomical structures and works on them in a 3D environment registered using fundamental imaging techniques, and thus AR augments the physical elements with virtual elements [1]. Integral imaging stands somewhere between stereoscopy and holography, both of which have their limitations. The devices required for integral imaging are simply a liquid-crystal display (LCD) and a lens in front of the LCD. The resulting 3D image is easily viewed by the user without the need for special eyewear [3]. When digital images are placed upon real images and alterations are done virtually, the results that are intended to be clinically achieved can be experienced pre-operatively [4]. This familiarizes the patient with the steps of treatment and its outcome. In the case of a smile makeover or complete mouth rehabilitation, before any dental procedure is performed, AR enables patients to get a glimpse of what they will look like once the treatment is undertaken by the dentist/dental surgeon. This helps reduce the number of visits and is time and money-saving for both the patient and the dentist, along with many other advantages associated with the use of AR/VR.

On the other hand, virtual reality makes use of technologically advanced and customized software to visualize a digital three-dimensional reality where the user's senses are stimulated using computer-generated feedback and sensations. Therefore, virtual reality allows the user to participate in virtual realities related to the physical reality, thus making virtual reality indistinguishable from physical reality. AR and VR differ from each other in several ways such as AR users can control their presence in the real world, but VR is system controlled. Secondly, the use of VR requires a headset device, but AR can be used with a smartphone. Moreover, VR can only enhance fictional reality, but AR can improve real and virtual worlds.

Virtual simulators have an edge over physical simulators as they allow the operator to go back in time and make amendments to the procedure that has been done. These virtual practice sessions improve the efficiency of the operating personnel and make the

outcome predictable [5]. According to the degree of immersion experienced by the user, virtual reality has three broad classifications: immersive virtual reality, which consists of a head-mounted display system, non-immersive virtual reality, which is based on a computer-aided (CAD) network, and semi-immersive virtual reality, based on a cave automatic virtual environment [1,6]. In immersive virtual reality, the operator is immersed in a virtual environment where they can interact with recorded 3D images/objects using a wearable device/headset and a pen-shaped manipulator. Eye movements and leap motion of the hands are detectable through this device [4]. However, in non-immersive virtual reality, the operator interacts with 3D simulations on a desktop computer displayed on a flat-screen 2D monitor using a mouse instead of a wearable device and manipulates the virtual images without being a part of a virtual scenario [4].

In regard to the sub-specialties of dentistry, so far, augmented reality and virtual reality have been mostly applied in the fields of dental implantology and orthognathic surgery. In augmented reality scenarios, the visual information gathered by registering anatomical structures and overlaying them on actual operative sites provides better guidance for surgical and other dental procedures. In some AR systems, 2D-projected computer graphics (CG) do not provide the viewer with the perception of depth at the surgical site, thus reducing the safety and accuracy of the procedure. Hence 3D images generated either via stereoscopy [3], integral imaging (or integral photography) and holographic display [7] are projected onto the site of the procedure, which enables the user to spatially visualize and perceive the depth of tissues from all locations, which enhances safety and accuracy. Furthermore, the contemporary Microsoft Hololens waveguide AR system, significantly advances the optical design of current AR display systems and may also be applied to a broad range of optical systems, including high-precision imaging, sensing, and advanced photonic devices [6].

The purpose of this review was to provide an update on the contemporary knowledge, to report on the ongoing progress of AR and VR in various fields of dental medicine and education, and to identify further research needs that will achieve their clinical translation.

## 2. Materials and Methods

For this review, an electronic search was performed via PubMed, Scopus, Web of Science, and Google Scholar using the terms "Augmented Reality", "Virtual Reality", "AR and VR", "AR and Dentistry", "VR and Dentistry", "Use of AR and VR in Dentistry", "AR/VR and dental education", for published research articles from the year 2011 to 2021. A secondary search was further performed after analyzing the list of articles that met the pre-determined inclusion criteria of the study. Three independent researchers S.F., A.M. and N.A., read the articles that were retrieved by the search engines, and studies that did not meet the inclusion criteria were excluded. Due to the heterogeneity of the design of the included studies, a systematic review and meta-analyses could not be performed. Therefore, a narrative review with the "best-evidence synthesis" [8] approach was carried out.

The pre-determined inclusion criteria for this study were as follows:

- Clinical trials
- Case-control studies
- Observational studies
- Studies where AR and VR have focused on dentistry
- Studies in the English Language

The studies that were excluded from this study were:

- Review articles
- Short communications
- Letters to Editors
- Studies in languages other than English

The data that were collected from the studies were extracted by the investigators and then noted under the following headings: "authors", "year of publication", "study

type", "sample size", "outcomes", and "field of dentistry". Any disagreement amongst the investigators was solved by discussion with a third reviewer, A.L.

## 3. Outcomes of Literature Search

The initial search of the databases resulted in a total of 101 articles. After analysis of the abstracts, title, and duplications of the studies, 81 articles were removed based on the inclusion criteria specified for this review. A total of 20 articles that fulfilled the inclusion and exclusion criteria were included in this review article.

The general characteristics of the studies included in this study have been described in Table 1. AR and VR have been successfully used in various fields of dentistry such as oral and maxillofacial surgery, orthodontics, endodontics, dental implantology, and dental public health. Dental anxiety is a problematic experience for both patients and dentists, especially pediatric patients. Such anxieties have been successfully reduced by the use of AR and VR whereby children were subjected to watching cartoons using virtual reality goggles, resulting in a decrease in heart rates and pain scores [9,10]. The use of AR and VR has further improved the surgical accuracy and duration of surgery along with the manual dexterity of surgeons who employ the use of AR and VR [11–14]. The development of AR devices has allowed users to combine all sorts of information and images that are converted into reality. Furthermore, orbital reconstruction and placement of the dental implants in the alveolar bones of patients require rigorous planning and surgical skills for optimal outcomes, which are further enhanced by the use of AR and VR [12,15].

The use of AR and VR has been introduced into operative dentistry residency training, which helps the residents to improve their confidence and knowledge by practicing the skills virtually before the implementation of their skills on patients [16]. Trauma is a common finding in dentistry, ranging from trauma affecting a single tooth to large facial fractures. Traumas and tumors are known to involve the orbits, so the introduction of the use of AR and VR in the reconstruction of the orbit has been used successfully [17]. Moreover, in the field of oral and maxillofacial surgery, the use of AR and VR in performing mandibular angle oblique split osteotomy has been studied, and improved and effective results were obtained [18]. The use of AR technology in oblique split osteotomy proved to be helpful for controlling maxillary translocation during orthognathic surgery.

Since CT scans form an integral part of treatment planning and the execution of the treatment of patients, AR and VR were introduced in CT-scan imaging for oral and maxillofacial surgery to provide images of higher quality [6]. Such CT-scan images provided better visualization for surgeons, of various anatomic structures in the area of the operative field. For a successful root canal treatment, the, detection of all canal orifices is crucial, so AR and VR use in endodontics has been explored with real-time detection of root canal orifices [19]. Since it is known that the identification of all root canal orifices is of prime importance for the success of root canal treatment, the use of AR and VR technology in endodontics can prove to be fruitful for endodontists.

For the correction of malocclusion in patients, at times, surgical corrections are required during their orthodontic treatment. So, AR and VR have been used in patients requiring surgical orthodontics where the guided placement of brackets for orthodontics correction was performed [20]. This method captured several sequences of patients' video to help improve the robustness, performance, and accuracy for more efficient orthodontic treatment. For students, to improve their knowledge and clinical skills, AR and VR have been introduced to further improve the level of education of students during their undergraduate years [21].

**Table 1.** Application of augmented reality and virtual reality in dentistry.

| Author ID and Year | Study Design | Sample Size | Outcomes | Field of Dentistry |
|---|---|---|---|---|
| Longkuan et al. [9] 2021 | Clinical Trial | 120 | The use of virtual reality resulted in decreased anxiety and pain in children | Pediatric Dentistry |
| Osama et al. [10] 2021 | Clinical Trial | 50 | Virtual reality is effective in reducing anxiety and pain | Pediatric Dentistry |
| Liu et al. [22] 2021 | Cross-sectional study | 5 | Augmented reality improved the efficiency and safety of craniofacial fibrous dysplasia recontouring | Oral and Maxillofacial Surgery |
| Lahti et al. [23] 2020 | Clinical Trial | 255 | Virtual reality resulted in a decrease in preoperative dental anxiety | Dental Public Health |
| Gujjar et al. [11] 2019 | Clinical Trial | 30 | Virtual reality was found to be effective in dental phobia treatment | Preventive Dentistry |
| Jiang et al. [12] 2018 | Clinical Trial | 12 | Improved accuracy and efficiency of virtual reality | Dental Implantology |
| Murugesan et al. [13] 2018 | Experimental Study | 23 | Expectable accuracy of VR | Oral and Maxillofacial Surgery |
| Pulijala et al. [14] 2018 | Clinical Trial | 98 | VR improves self-confidence and knowledge amongst surgical residents | Oral and Maxillofacial Surgery |
| Schreurs et al. [15] 2018 | Pilot Study | 1 | A novel concept of orbital reconstruction | Oral and Maxillofacial Surgery |
| Llena et al. [16] 2018 | Case-control | 41 | AR resulted in improved knowledge and skills | Operative Dentistry |
| Won YJ et al. [17] 2017 | Descriptive | 1 | Use of VR in inferior alveolar nerve block | Oral and Maxillofacial Surgery |
| Zhu et al. [18] 2017 | Clinical Trial | 20 | AR resulted in improved and effective mandibular angle oblique split osteotomy | Oral and Maxillofacial surgery |
| Bruellmann et al. [19] 2013 | In vitro study | 126 | Real-time detection of root canal orifices | Endodontics |
| Aichert et al. [20] 2012 | Experimental study | 1 | Use of AR in orthodontics procedure | Orthodontics |
| Bogdan et al. [21] 2011 | Descriptive study | Not mentioned | Use of VR for dental education of students | Dental Education |
| Wang et al. [24] 2017 | Clinical Trial | 2 | VR can be integrated into OMFS | Oral and Maxillofacial Surgery |
| Suenaga et al. [6] 2015 | Clinical Trial | 1 | The use of AR improves 3D-CT images with higher accuracy | Oral and Maxillofacial Surgery |
| Qu M et al. [25] 2015 | Clinical Trial | 20 | AR enhances intraoperative distraction osteogenesis | Oral and Maxillofacial Surgery |
| Badiali et al. [26] 2014 | Experimental Study | 1 | AR enhances surgical procedures | Oral and Maxillofacial Surgery |
| Lin et al. [27] 2013 | In vitro study | 40 | AR-enhanced implant placement | Oral Implantology |

## 4. Applications of AR and VR in Dentistry

### 4.1. AR and VR in Dental Specialties

AR and VR have their use in a variety of specialties of dentistry, as presented in Figure 1. AR and VR technologies have a promising role in the field of dentistry. The details of its application in oral health are described below.

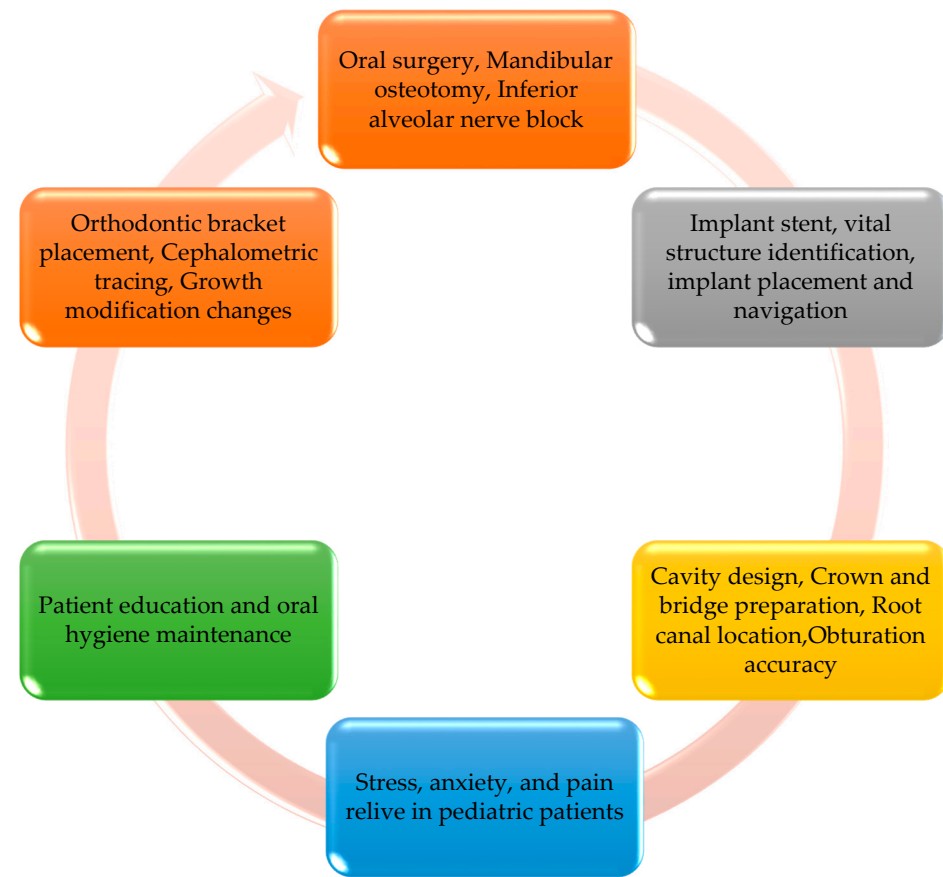

**Figure 1.** Use of virtual /augmented reality in dentistry.

### *4.2. AR and VR in Dental Education/Training*

Traditionally, the training of dental students has been conducted on phantom heads and teeth to practice and improve their clinical skills before performing treatments on patients. These simulators allow teachers to demonstrate the treatment techniques and aim to enhance the manual dexterity of the students. The use of such simulators makes it mandatory for the students to give their teachers constant feedback about their progress before moving to different treatment techniques.

In the delivery of dental education and training, instructors can help students and PG residents to achieve precision in pre-clinical/clinical skills, respectively, with 3D real-time digital simulations as well [1,9]. Learning and retaining the anatomy of the head and neck is a crucial part of dental education. Lectures and 2D images from textbooks of anatomy have been used for teaching this subject to students [10]. In this traditional mode of teaching, cadaveric skulls are commonly employed, and students are instructed through these. However, the visualization of all associated muscular, neural, vascular, and other structures is quite challenging for students and it is not fully effective either [22].

In the midst of the COVID-19 pandemic, students suffered due to limited learning opportunities and the ability to enhance their pre-clinical skills as students could not train at their universities without the instructor's direct supervision. Therefore, helping these students to learn while controlling the spread of COVID-19 has been the most challenging aspect for the universities. Amidst the quest for better teaching methodologies, one group described an innovative technique to teach clinically relevant anatomy to students of dentistry that replaces cadavers with dissected and sliced plastinated specimens [8]. However, one recent study incorporated digital tools and a 3D augmented curriculum, which is an interactive 3D experience that combines the view of the actual world with elements generated by computers along with traditional teaching methods (lectures and cadavers). The study demonstrated improved understanding and a positive influence on

the retention of anatomical knowledge in students as compared to the control group (taught via textbooks/2D images) [23].

To further increase the knowledge of dental students, virtual reality allows the students to watch oral treatments as a direct participant. Moreover, students can also try to reproduce the oral treatment scenario under the guidance of professionals.

### 4.3. AR and VR in Oral and Maxillofacial Surgery

Oral pathologies such as oral squamous cell carcinoma, cleft lip and palate, and congenital abnormalities are common findings that are treated by oral and maxillofacial surgeons. Many of these pathologies are treated by surgeons using their manual dexterity along with their years of experience. In recent years, many technological advancements have taken place in the field of surgery such as the introduction of augmented and virtual reality [24].

The application of AR and VR technology has been explored in many surgical fields of medical science, such as laparoscopic surgery, neurosurgery, and plastic surgery. In dentistry, the use of AR in oral and maxillofacial surgery has focused on the placement of dental implants, craniofacial surgery, and orthognathic surgery. The use of AR technology allows the users to combine information and images to bring them to reality.

The training period of surgical residents is a crucial phase where the residents learn and practice their surgical skills on different simulators before actually performing on patients. Pulijala et al. [14] evaluated the effectiveness of virtual reality in surgical training and found that many of the surgeons were not confident about performing the surgeries. With the introduction and implementation of VR technologies, this has resulted in surgical residents improving their knowledge and confidence whilst performing the surgeries. VR is an additional tool that has immense importance in further enhancing the skills already possessed by surgeons to achieve optimal outcomes that boost confidence amongst surgeons.

Inferior alveolar block anesthesia is one of the most fundamental anesthesia used in dentistry to operate on the mandibular teeth for procedures such as root canal treatment, extractions, dental fillings, and complex surgeries. Many factors have been associated with the failure of inferior alveolar block anesthesia such as poor technique, and anatomical variations. In a study by Won et al. [17], AR was used for inferior alveolar block anesthesia and it was concluded the use of AR in this block anesthesia can improve the effectiveness when block anesthesia is used alone. AR helps in improving the precision and accuracy of using block anesthesia as the images that are directly generated from the patient help in converting those images to reality.

One of the most widely studied applications of AR and VR in dentistry is orthognathic surgery. The prime advantage of using AR-guided navigation tools is that it provides real surgical images and virtual surgical plans to guide through the treatment plan. One of the important surgical procedures in orthognathic surgery is mandibular angle split osteotomy. Mandibular angle split osteotomy is a cosmetic surgical procedure that aims to improve the prominent mandibular angle, thereby improving the aesthetics of the patient.

In their study, Zhu et al. [18] used AR for mandibular angle split osteotomy and found that the use of AR enhances the translocation of the maxilla in orthognathic surgery. The use of such AR systems allows surgeons to operate real-time streaming video images that allow them to plan the surgery and study the anatomical structures of the patient, thus enhancing the accuracy of such orthognathic surgical procedures.

The use of AR has also been studied in distraction osteogenesis. In a study by Qu et al. [25], patients suffering from hemifacial microsomia were treated with an intraoral distractor using AR and it was found that AR was more accurate in proper positioning of the osteotomy planes as compared to the conventional methods. Additionally, one study explored the use of AR systems where images were overlaid, which allowed surgeons to observe and follow the virtual surgical plans to reposition the bones of the patient after performing maxillary osteotomies [5]. Additionally, to further enhance the surgical skills, VR technology replicated different functions such as drilling, place fixation, and bone sawing with the help of hepatic force feedback [28,29].

So, the importance of VR and AR in oral and maxillofacial surgery can be appreciated given the improved accuracy of the surgery performed, the decrease in the chance of errors by the surgeons, and the unlimited number of training sessions available to surgeons. Therefore, VR and AR as an adjunct can prove to be useful tools for surgeons.

### 4.4. AR and VR in Paediatric Dentistry

Pediatric dentistry is one of the most challenging specialties of dentistry as the commonest factor determining the treatment outcomes in such patients is their compliance. To improve patient cooperation and compliance, different tools have been used ranging from the armamentarium of the dentist, such as behavior modification, and pharmacological interventions. Pediatric patients who are visiting the dental practice often present with tremendous amounts of anxiety as most of the time it is their first interaction with dentists [30]. So, managing the anxieties and behavior of children is one of the crucial factors in patient management.

To decrease the levels of anxiety and stress of such patients, virtual reality is one of the innovative tools that have been discussed in the literature, although to a limited extent. Different techniques to manage anxiety include in vivo exposure therapy (IVET) and virtual reality exposure therapy (VRET). IVET consists of the direct confrontation of patient's fear to reduce their anxiety levels and this method has been categorized as a gold standard method. VRET is a recent technique that consists of computer-generated images for patients where the simulation makes the patient experience their fears without facing them in reality, thereby helping them to reduce their anxiety [31]. In a study by Ran et al. [9], the effect of virtual reality on the behavioral management of children was studied, where it was concluded that the average anxiety and behavioral scores of the patients with virtual reality was significantly reduced as compared to the control group. Since virtual reality makes use of interactive and creative audiovisual representations of information that is attractive to children, such a method of anxiety reduction can be beneficial.

Another study by Osama et al. [10] assessed the effect of virtual reality on the pain and anxiety of pediatric patients during infiltration anesthesia, where it was found that virtual reality was effective in reducing the anxiety and stress of these patients. Since the patients experience the entire scenario virtually before the actual procedure commences, this helps patients to understand the treatment and face their fears. So, one of the most important factors that determine the treatment outcomes in patients in this age group is the anxiety and stress associated with visiting the dental practice. Virtual reality can help to provide an artificial environment that is more relaxing for the patients, which can help them forget their fears.

### 4.5. AR and VR in Dental Implantology

Many advances have been made in dentistry in regard to the replacement of missing teeth or teeth in the mandibular and maxillary arches of patients such as removable dentures, fixed dental prostheses, and dental implants. Dental implants have emerged as a suitable and preferable choice for many patients because of their high success rate and the long-term benefits associated with them [27,32]. The introduction of AR technology in dental implants has significantly improved many procedures associated with the placement of implants. Initially, the AR surgical navigation technology was used to place implants using the retinal imaging display as the surgeon keeps his visual on the operative area, avoiding the need for the surgeon to turn away [33].

In a study by Jiang et al. [12], it was found that the use of augmented reality resulted in higher accuracy and applicability for guided placement of dental implants as compared to the traditional two-dimensional navigational method. Such AR navigation systems help the surgeon to concentrate only on the site of the implant placement, thereby providing only useful information to the surgeon, which eventually reduces the cost and time of the procedure [34]. It is of prime importance that the location of the implants should be as

accurate as possible because negligence in this step of implant placement could be one of the factors responsible for future implant failure.

Moreover, one study used augmented reality-based dental implant placement to evaluate the virtual placement of the implant as compared to the actual prepared implant site created [27]. In this study, it was found that the use of augmented reality resulted in a significantly decreased deviation in implant placement from the actual planned site. Since AR navigation-based systems help the surgeon to accurately locate and place the dental implants, the accuracy of such procedures thereby increases as compared to the traditional methods.

The placement of the dental implant is a surgical procedure, so with the help of VR technology, patients have an opportunity to get a detailed explanation of the treatment and what they will experience while the treatment is being performed. Complete information given to the patients with the help of VR makes them better prepared mentally.

### 4.6. AR and VR in Restorative Dentistry and Endodontics

Restorative dentistry and endodontics are some of the most challenging and exhausting fields of dentistry. In order to treat diseases such as dental caries, pulpitis, and dental abscess, knowledge and clinical skills are required to become a proficient clinician [35]. A high caries rate is prevalent throughout the globe, which mandates a visit the dentist and may require dental fillings or even root canal treatment as well. These treatments are performed to save a tooth that might require extraction if not managed in a timely fashion. Many factors are associated with failed dental fillings and endodontic treatments such as isolation, poor technique, smoking, underfilled canals, and missed canals [19].

Conventionally, students are first trained in the laboratory on a phantom head where the mannequin mimics the patient, which allows the students to work as if they are working on an actual patient as in a clinical setup [36]. Procedures such as cavity preparation are practiced on these mannequins by the students. In a study by Llena et al. [16], cavity preparation using AR technology was studied. In this study, it was found that the participants that used AR technology showed an improvement in their knowledge and skills. With the help of AR, realistic simulations can be delivered to the students in order to practice and improve their clinical skills without the need for live test subjects.

To overcome the chances of failure in endodontic treatments in patients, augmented reality has been used in endodontics for the reliable detection of root canals. In a study by Bruellmann et al. [19], augmented reality was used to detect root canals where it was found that overall higher sensitivity was noted in detecting root canals in molars and premolars. The success rate of root canal treatment directly depends on identifying all of the root canals as any missed canal has pulp remnants that might trigger pain for the patient [37,38].

## 5. Clinical Implications of Haptic Feedback in AR/VR

Haptic feedback refers to the experience of physical resistance or vibration upon manipulating oral structures with a hand-held VR tool. Augmented or virtual reality combined with haptic feedback provides a realistic environment for dental operations. Dental procedures mostly require a bi-manual technique of instrumentation inside the oral cavity. Simulating this with haptic feedback is a challenge for software developers in the healthcare industry. Simulating the force response of oral structures and contact of rigid and deformable structures is necessary to carry out accurate haptic interactions in a virtual environment [39].

Practicing on virtual patients with real-time haptic feedback allows dental surgeons or students to perform routine and complex procedures quickly and with efficacy, although some studies have pointed out serious limitations for users working with haptics [40].

## 6. Whether to Embrace AR/VR Systems or Not?

In dentistry, the potential impact of the AR and VR technology is tremendous because all the natural occurrences that take place when a procedure is performed on a real patient can be simulated as well, e.g., during training for a root canal procedure, bleeding upon

perforation of the pulpal floor can be experienced by immersion in the virtual environment (VR goggles plus haptic peripherals) [41]. Given the recent pandemic, a keen focus on the education of dental students utilizing cutting-edge technology is warranted to enhance clinical skills [42]. Research in the field of dentistry can also employ AR/VR systems to investigate study participants remotely in a 4D environment (Hologram); this would reduce study-related expenditures as well and conveniently tackle the issue of ethical sensitivity in some studies. With virtual reality simulators, the wastage of materials is reduced as students/dentists do not need to practice on artificial teeth and models [43].

Therefore, virtual reality applications are also beneficial in preserving the natural environment. The benefits of AR/VR-based dentistry and conventional dentistry are shown in Table 2. All such results have led to a surge in interest in these technologies. However, the visual fatigue caused by a stereoscopic view in children and adults is concerning [3]. Pediatric and geriatric patients in dental care settings could be negatively affected by frequent and repeated exposures to AR/VR environments [9,10,38,44,45]. This needs to be evaluated through repeated randomized control trials (RCTs) of dental procedure-based simulations in patients of different age groups. Based on cognitive load theory (CLT), explorative studies with appropriate and validated questionnaires given to large numbers of participants should be conducted to validate the reports of cybersickness and sensory overload associated with AR/VR-based training and treatments.

**Table 2.** Comparison of the benefits offered in AR/VR-based dentistry and traditional dentistry.

| Parameters | Traditional Dentistry | AR-Based Dentistry | VR-Based Dentistry |
|---|---|---|---|
| Alleviation of Dental Anxiety Pre-Operatively | Achieved Verbally OR via Sedation/ Psychotherapy (Practiced Commonly Worldwide) | Smartphone/Tablet-Based AR Exposure Apps (restricted to Laboratories and Experiments) | VR Headset soothes patient with a tranquil virtual environment (Integration in Limited number of Dental Offices) |
| Pain Management | Pharmacotherapy/ prophylaxis-based pain management | AR simulators enhance pain threshold | Pain intensity is perceived artificially through a virtual simulation |
| Training Opportunity | Not effective for complex cases; poses risk for real patients | Dental Training units featuring AR-based systems; complex cases practiced safely | Dental training units featuring haptic technology: complex cases practiced safely and with precision with haptic feedback |
| Patients' Accurately Understand Dental Procedures | Not Reliable | Fully Reliable | Fully Reliable |
| Remote Equipment Repairs | Not possible hence procedures are halted until a technician repairs in-person | Unexplored | Possible hence faulty equipment repaired easily with technician's guidance remotely |
| Patients' Symptoms Experienced by Dentist First-hand | The dentist cannot relate accurately | Unexplored | The dentist is fully empathetic as one can experience a particular symptom virtually |
| 360-degree View of the Operatory Experienced Remotely | Possible but does not provide a realistic experience | Provides incredibly realistic experience | Provides fully immersive virtual views |

When there is an equipment failure in dental clinics, it might take a long period of time for the technician to arrive and perform the repair. However, with the help of AR, a repair can be performed quickly by the use of a head-mounted display where the dentist coordinates with the technician remotely and repairs the equipment themselves [46–49]. However, since the cost of AR and VR technology is high for the majority of dental clinics, the benefits of such technology are yet to be further explored.

## 7. Future Recommendations

Virtual reality creates a learning opportunity for dental surgeons to practice safe and efficient dentistry with a constant feedback mechanism. The current ongoing pandemic has demonstrated the importance of infection control in all healthcare environments now more than ever. Considering this, augmented and virtual reality-based dentistry may be even more applicable post-pandemic than before. It cannot be stressed enough that educators and clinicians need to consider the pros and cons before investing in the armamentarium required for AR/VR applications. Collaboration between experts working in the fields of Medical Informatics and Public Health Informatics with clinicians to translate AR and VR-based treatment planning and procedures into routine clinical work is an active area for future research. Moreover, further exploration is needed to evaluate the usefulness of AR/VR in dental education for different dental practitioners to practice different treatments independently.

## 8. Conclusions

The field of dentistry is advancing at a rapid pace, in this regard, new technologies are being developed that a dentist could benefit from; these benefits include better visualization potential, reduced operative time, better patient consultation and promising treatment outcomes. The use of AR and VR has been studied in different fields of dentistry, but there were limited studies assessing its use. In this review, augmented reality and virtual reality have been found to be beneficial tools for clinical practice in the field of oral maxillofacial surgery, preventive dentistry, endodontics, and orthodontics. Clinicians can use virtual reality technology to show their patients the expected outcomes before they even undergo any procedures. AR and VR also have a potential role in dental education through enhancing the learning experience for students during their pre-clinical education and training. Additionally, AR and VR can also be implemented to overcome dental phobia, which is commonly experienced by pediatric patients. Future studies should focus on forming technological standards with high-quality data and developing scientifically proven AR/VR gadgets for dental practice.

**Author Contributions:** Conceptualization, A.L., S.F., N.A., M.K.A., A.M., G.D., S.S. and A.A.G.K.; methodology, A.L., S.F., N.A. and A.M.; software, N.A., G.D., M.K.A. and A.M.; validation, M.K.A., N.A. and A.L.; formal analysis, A.L. and N.A.; investigation, A.L., S.S. and A.M.; resources, M.K.A., A.A.G.K. and S.S.; data curation, A.L. and N.A.; writing—original draft preparation, A.L., S.F., N.A., M.K.A., A.M., S.S. and A.A.G.K.; writing—review and editing, N.A., M.K.A. and A.M.; visualization, A.L., M.K.A. and N.A.; supervision. A.M. and N.A.; project administration, A.L., N.A. and S.F.; funding acquisition, M.K.A., G.D., S.S. and A.A.G.K. All authors have read and agreed to the published version of the manuscript.

**Funding:** This research was funded by the Deanship of Scientific Research, King Khalid University, Abha-Asir, Kingdom of Saudi Arabia under the Small Research Group, with the grant number, RGP.1/336/42.

**Institutional Review Board Statement:** Not applicable.

**Informed Consent Statement:** Not applicable.

**Data Availability Statement:** The data presented in this study are available on request from the corresponding author.

**Acknowledgments:** Authors thankfully acknowledge the Deanship of Scientific Research, King Khalid University, Abha-Asir, Kingdom of Saudi Arabia for funding this research work under the Small Research Group, with the grant number, RGP.1/336/42.

**Conflicts of Interest:** The authors declare no conflict of interest.

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
