# Peer review of "Augmented Reality and Virtual Reality in Dentistry: Highlights from the Current Research"

_applsci, doi:10.3390/app12083719_

Round 1

Reviewer 1 Report

I think the author has done well to address some of the issues raised.

However, the abstract is still not structured so well to provide strong summary of this study. For example, the argument on why this study is necessary and important is not clear and the findings from this study were missing in the abstract.

I understand that the authors have used narrative review as their methodology. Even though I have my reservations for this methodology, authors could have made a strong case why they chose it over other methods of review such as a systematic literature review. I will encourage the authors to reference this previous study (https://familymedicine.med.wayne.edu/mph/project/green_2006_narrative_literature_reviews.pdf) that provides useful information on how to conduct a narrative review and update their abstract.

Author Response

Point to point author team response to reviewer comments

Thank you for reviewing our manuscript. The corrections recommended by the respected reviewers are addressed in different sections of the manuscript. The corrections are further highlighted with distinct colors for clarity. The detailed response to the reviewer’s comment is described  below:

Reviewer 1 comments:

I think the author has done well to address some of the issues raised.

However, the abstract is still not structured so well to provide strong summary of this study. For example, the argument on why this study is necessary and important is not clear and the findings from this study were missing in the abstract.

Authors response: Thank you, for the appreciation. The abstract section is corrected and revised, Page number 1, main document.

I understand that the authors have used narrative review as their methodology. Even though I have my reservations for this methodology, authors could have made a strong case why they chose it over other methods of review such as a systematic literature review. I will encourage the authors to reference this previous study (https://familymedicine.med.wayne.edu/mph/project/green_2006_narrative_literature_reviews.pdf) that provides useful information on how to conduct a narrative review and update their abstract.

Authors response: Thank you, Due to the heterogeneity of the design of included studies, systematic review and meta-analyses could not be performed. Moreover, recent systematic reviews in the last 5 years were available in the databases, focusing on a specific study type…i.e., trials, etc.

Moreover, the article recommended is cited in the current manuscript text, reference number 8.

Reviewer 2 Report

The authors have made significant improvements to the manuscript by previous suggested revisions. There are a still a few more revisions that could further help the quality of the manuscript as follows:

Introduction

Provide a sentence explaining the purposes of this review in the last paragraph of the Introduction.

Outcomes of Review

Table 1 - include periods after et al. in Column 1 all reference citations

References

Reference 3 - Line 470-72 - remove large spaces between words

Place periods after all initials of authors (for all references)

Author Response

Point to point author team response to a reviewer comment

Thank you for reviewing our manuscript. The corrections recommended by the respected reviewer have been addressed in different sections of the manuscript. The corrections are further highlighted with distinct color for clarity. The detailed response to reviewer’s comment is described  below:

Reviewer 2 comments

The authors have made significant improvements to the manuscript by previous suggested revisions. There are a still a few more revisions that could further help the quality of the manuscript as follows:

Introduction

Provide a sentence explaining the purposes of this review in the last paragraph of the Introduction.

Authors response: Thank you, I corrected and revised the introduction section, the purpose of the study is added to the last paragraph, page no 3, the main document

Outcomes of Review

Table 1 - include periods after et al. in Column 1 all reference citations

Authors response: Thank you, the I corrected and revised the column 1 of table number 1.

References

Reference 3 - Line 470-72 - remove large spaces between words

Authors response: Thank you, the correction is carried out in the references section throughout.

Place periods after all initials of authors (for all references)

Authors response: Thank you, we have revised and corrected the references

Reviewer 3 Report

This is a good manuscript discussing and summarizing the development of augmented reality and virtual reality applied in dental medicine and dental education. The manuscript is well organized including the research studying methods, the well-done table summary, the application in dental specialties, dental education, oral and maxillofacial surgery, pediatric dentistry, dental implantology, restorative dentistry and endodontics. The manuscript had done plenty of the search works which is valuable for later researchers to quickly move to the frontier of related research area. 
However, there are some questions/issues in the manuscripts, I suggest publishing this manuscript after the minor change/modification. 
1. The paper format issue needs to have more check. For example, the yellow highlighted words, the non aligned reference format. 
2. The reference needs to be clearer. For example,  when I read the manuscript and tried to follow the ref 8 to know more about the holographic display, I cannot get it since the it links to a book without certain pages. 
3. When the author mentioned the different AR techniques in line 119 and 120, looks there are more advanced AR techniques/devices currently applied in medical area that haven’t been mentioned, like Microsoft Hololens waveguide AR. 

Author Response

Point to point author team response to reviewer comments

Thank you for reviewing our manuscript. The corrections recommended by the respected reviewer have been addressed in different sections of the manuscript. The corrections are further highlighted with distinct color for clarity. The detailed response to the reviewer’s comment is described  below:

Reviewer 3 comments

This is a good manuscript discussing and summarizing the development of augmented reality and virtual reality applied in dental medicine and dental education. The manuscript is well organized including the research studying methods, the well-done table summary, the application in dental specialties, dental education, oral and maxillofacial surgery, pediatric dentistry, dental implantology, restorative dentistry and endodontics. The manuscript had done plenty of the search works which is valuable for later researchers to quickly move to the frontier of related research area.

However, there are some questions/issues in the manuscripts, I suggest publishing this manuscript after the minor change/modification.

Authors response: Thank you, the author’s team is grateful for the honest opinion and recommendation of suggestions to improve our manuscript. the manuscript is re-evaluated for corrections needed in different sections, The response to comments is given below.

  1. The paper format issue needs to have more check. For example, the yellow highlighted words, the non aligned reference format.

Authors response: Thank you, the format of the manuscript re checked, and revised by the authors team.

  1. The reference needs to be clearer. For example, when I read the manuscript and tried to follow the ref 8 to know more about the holographic display, I cannot get it since the it links to a book without certain pages.

Authors response: Thank you, the references are revised and checked for accuracy.

  1. When the author mentioned the different AR techniques in line 119 and 120, looks there are more advanced AR techniques/devices currently applied in medical area that haven’t been mentioned, like Microsoft Hololens waveguide AR.

Authors response: Thank you, the text is amended in introduction section, we have added the following text in the 7th paragraph, main document.

Furthermore, the contemporary Microsoft Hololens waveguide AR system, significantly advances the optical design of present AR display systems and also open new applications to a broad range of optical systems, including high-precision imaging, sensing, and advanced photonic devices.

This manuscript is a resubmission of an earlier submission. The following is a list of the peer review reports and author responses from that submission.

Round 1

Reviewer 1 Report

Well written! Acceptable as per me

Author Response

Point to point authors response to reviewer comments

Thank you for reviewing our manuscript. The corrections recommended by the respected reviewers are addressed in different sections of the manuscript. The track change is on in the manuscript. The corrections are further highlighted with distinct color for clarity. The detailed response to reviewer comments is described  below:

Reviewer 1

Well written! Acceptable as per me

Author response: Thank you, we appreciate your valuable comment, the author’s team is grateful.

Reviewer 2 Report

The authors have provided an review of the potential applications of augmented and virtual reality in dentistry. There following is a list of suggested revisions to significantly improve the quality and clarity of the manuscript.

Title - please rephrase to improve clarity (as follows)

Augmented and Virtual Reality in Dentistry: Highlights from Current Research

Introduction - section 1

There are only 2 references in the Introduction!. At least 25% of the references (literature review of past & present application) should be in the Introduction. Please also provide more very recent references 2018-2021 in the Introduction.

There should be no Figures in the Introduction. Start figures in section 2 and beyond.

Figure 2 - blue shadow highlighting not necessary (distracting)

Figure 3 - blue rectangle shadow not necessary (distracting)

Table 1 - p.5

Make column Titles bold

Place periods after et al., then a space before [ref#] for each reference

Table 2 - this table should be made into a figure. The text resolution  is very low and fuzzy, please increase resolution to at least 300 dpi.

Place periods after et al. and add reference numbers in the following cases:

L 186

L 204

Table 3 - column 3, add reference numbers (replace years with ref. no.)

          [for Zimmer 2021, and Mladenovic 2020]

References

Auto Reference numbering template was not used (with built-in margins), thus 2nd and 3rd lines come back to the margins (improper formatting)

All name abbreviation initials should have a period after each letter.

Place commas after last names and semicolons after each name initials (between authors).

Insert hyperlinks after each website URL (by selecting web address, then right-click and select add hyperlink)

Author Response

Point to point authors response to reviewer comments

Thank you for reviewing our manuscript. The corrections recommended by the respected reviewers are addressed in different sections of the manuscript. The track change is on in the manuscript. The corrections are further highlighted with distinct color for clarity. The detailed response to reviewer comments is described  below:

Reviewer 2

The authors have provided a review of the potential applications of augmented and virtual reality in dentistry. There following is a list of suggested revisions to significantly improve the quality and clarity of the manuscript.

Title - please rephrase to improve clarity (as follows)

Augmented and Virtual Reality in Dentistry: Highlights from Current Research

Author response: Thank you, the title is corrected as per valuable comments by the reviewer.

Introduction - section 1

There are only 2 references in the Introduction! at least 25% of the references (literature review of past & present application) should be in the Introduction. Please also provide more very recent references 2018-2021 in the Introduction.

Author response: Thank you, the correction is carried out. References are added in different subheadings of the introduction section. Page number

There should be no Figures in the Introduction. Start figures in section 2 and beyond.

Author response: Thank you, corrected.

Figure 2 - blue shadow highlighting not necessary (distracting)

Author response: Thank you, corrected.

Figure 3 - blue rectangle shadow not necessary (distracting)

Author response: Thank you, corrected.

Table 1 - p.5

Make column Titles bold

Author response: Thank you, corrected.

Place periods after et al., then a space before [ref#] for each reference

Author response: Thank you, corrected.

Table 2 - this table should be made into a figure. The text resolution  is very low and fuzzy, please increase resolution to at least 300 dpi.

Author response: Thank you, corrected. Table 2 has been converted into figure with resolution of 300 dpi.

Place periods after et al. and add reference numbers in the following cases:

L 186

L 204

Author response: Thank you, corrected.

Table 3 - column 3, add reference numbers (replace years with ref. no.)

          [for Zimmer 2021, and Mladenovic 2020]

Author response: Thank you, corrected.

References

Auto Reference numbering template was not used (with built-in margins), thus 2nd and 3rd lines come back to the margins (improper formatting)

Author response: Thank you, corrected.

All name abbreviation initials should have a period after each letter.

Author response: Thank you, corrected.

Place commas after last names and semicolons after each name initials (between authors).

Author response: Thank you, corrected.

Insert hyperlinks after each website URL (by selecting web address, then right-click and select add hyperlink)

Author response: Thank you, corrected.

Reviewer 3 Report

I consider  English language and style are required.

It could be better to try doing a systematic review of this interesting topic.

They should reduce the manuscript extension.

Author Response

Point to point authors response to reviewer comments

Thank you for reviewing our manuscript. The corrections recommended by the respected reviewers are addressed in different sections of the manuscript. The track change is on in the manuscript. The corrections are further highlighted with distinct color for clarity. The detailed response to reviewer comments is described  below:

Reviewer 3

I consider  English language and style are required.

Author response: Thank you, corrected at various sections of the manuscript and highlighted for clarity.

It could be better to try doing a systematic review of this interesting topic.

Author response: Thank you. The author team appreciate the suggestion, since a focused question was not followed and no ambiguity or confusion was found in the literature regarding AR and VR application in various dental fields, the authors team decided to do a narrative review.

They should reduce the manuscript extension.

Author response: Thank you, the correction is carried out, as per relevance the text is edited in the manuscript.

Reviewer 4 Report

This review aims to summarize some of the uses and advantages of adopting VR technology in dental practices and education. I appreciate the authors’ effort, but the review right now is just a list of other researchers’ results without any conceptual analysis (e.g., why are VR useful, what are the conditions to make it useful? When will it not be useful?). As such, this manuscript in its current form is too superficial and does not have the value of an adequate review paper. My detailed comments below.

  1. All of the figures in this manuscript can be eliminated as they are simply lists of things. They can be listed in text, or in the form of tables, but there is no need to represent the concepts/uses in spatial format unless there’s some conceptual connections that need to be depicted spatially (which I see none). As of now they are just words with clip arts, which is unnecessary.
  2. What would be useful for the figures would be some sample pictures of what the users might be seeing in VR, and, importantly, pictorially depict why that might be advantageous over a traditional 2D display.
  3. Figure 2 shows 3 categories but the text says 2.
  4. Line 125, what is the 3D augmented curriculum here? What additional information is offered? What are the students seeing that is helping them? The authors lists a lot of “what”, but there is no “how” or “why” whatsoever. This is a constant problem throughout the entire manuscript, the contents right now are too superficial, please offer more in-depth information that not only lists out the “what”, but why is VR helpful, and the how (learning mechanisms, additional information) and when (when will VR not be helpful?). Line 226 to 247 touches upon this idea but is not enough.
  5. Line 134 to 160, Line 174, again same problem, no “why”, just listing the superficial findings from other studies. Beyond listing the results, a review is supposed to organize the concepts for the readers.

Author Response

Point to point authors response to reviewer comments

Thank you for reviewing our manuscript. The corrections recommended by the respected reviewers are addressed in different sections of the manuscript. The track change is on in the manuscript. The corrections are further highlighted with distinct color for clarity. The detailed response to reviewer comments is described  below:

Reviewer 4

This review aims to summarize some of the uses and advantages of adopting VR technology in dental practices and education. I appreciate the authors’ effort, but the review right now is just a list of other researchers’ results without any conceptual analysis (e.g., why are VR useful, what are the conditions to make it useful? When will it not be useful?). As such, this manuscript in its current form is too superficial and does not have the value of an adequate review paper. My detailed comments are below.

  1. All of the figures in this manuscript can be eliminated as they are simply lists of things. They can be listed in text, or in the form of tables, but there is no need to represent the concepts/uses in spatial format unless there’s some conceptual connections that need to be depicted spatially (which I see none). As of now they are just words with clip arts, which is unnecessary.

Author response: Thank you. We appreciate the respected reviewer’s comment. The reason for mentioning figures in the manuscript is to provide; the quickest way to communicate large amounts of complex information used regarding the topic. The author team also believes that many readers will only look at our display items without reading the main text of the manuscript. If the respected reviewer allows us to keep the figures in the paper, it will improve the face of our manuscript and facilitate readers.

  1. What would be useful for the figures would be some sample pictures of what the users might be seeing in VR, and importantly, pictorially depict why that might be advantageous over a traditional 2D display.

Author response: Thank you, we are pleased by the comment, indeed it is important to highlight the advantages of cotemporary AR and VR, to conventional 2D display. In this paper Table number 02 and figure 03 is focusing on advantages of AR and VR over traditional methods

  1. Figure 2 shows 3 categories, but the text says 2.

Author response: Thank you, corrected. Line number 186, the figure is number is changed to 1 in the text due to editing.

  1. Line 125, what is the 3D augmented curriculum here? What additional information is offered? What are the students seeing that is helping them? The authors lists a lot of “what”, but there is no “how” or “why” whatsoever. This is a constant problem throughout the entire manuscript, the contents right now are too superficial, please offer more in-depth information that not only lists out the “what”, but why is VR helpful, and the how (learning mechanisms, additional information) and when (when will VR not be helpful?). Line 226 to 247 touches upon this idea but is not enough.

Author response: Thank you, the correction is carried out in corrected. Page number 6 and 12, Line number 134-136, and 256-260, introduction and discussion sections, main document.

  1. Line 134 to 160, Line 174, again same problem, no “why”, just listing the superficial findings from other studies. Beyond listing the results, a review is supposed to organize the concepts for the readers.

Author response: Thank you, we apologize for the mistake, the correction is carried out in the introduction section.

Reviewer 5 Report

Title

Why did the authors use the term “Augmented Reality / Virtual Reality” instead of “Augmented Reality and Virtual Reality”? I found the use of “/” problematic when the main text contains “Augmented Reality and Virtual Reality “or even “AR and VR”. I think the title should be “Augmented Reality and Virtual Reality in Dentistry: Highlights from Current Research”. This should be fixed in all parts of the manuscript in order to maintain consistency.

 Abstract:

Should contain a bit of the research focus should be provided in the abstract. Just mentioning that “we brief about the development and progressive increase in applications of VR and AR technology in various areas of dental medicine and dental education” is not enough for readers to gain knowledge about the study. 

In addition, the authors should include a few sentences in the abstract that highlights the methodology, interesting findings, and implication of the study. If the authors are concerned about the length of the abstract after these adjustments, then, the leading sentences should be removed and taken to the introduction section.

 Methods and presentation:

I really like the presentation of the authors and enjoyed reading the different sections. However, I’m wondering about the style of the presentation. I was expecting to see a structure that showcased the research questions, methodology, results, discussion, and conclusion. While readers may take away some knowledge regarding the overview of AR and VR in dental medicine, it is not clear about the science of the study and what specific question is addressed. Unless these issues are addressed, the manuscript may look more like a piece of report than a scientific paper

Author Response

Point to point authors response to reviewer comments

Thank you for reviewing our manuscript. The corrections recommended by the respected reviewers are addressed in different sections of the manuscript. The track change is on in the manuscript. The corrections are further highlighted with distinct color for clarity. The detailed response to reviewer comments is described  below:

Reviewer 5

Title

Why did the authors use the term “Augmented Reality / Virtual Reality” instead of “Augmented Reality and Virtual Reality”? I found the use of “/” problematic when the main text contains “Augmented Reality and Virtual Reality “or even “AR and VR”. I think the title should be “Augmented Reality and Virtual Reality in Dentistry: Highlights from Current Research”. This should be fixed in all parts of the manuscript in order to maintain consistency.

Author response: Thank you, the title of manuscript is revised as per valuable comments of reviewer, moreover the term AR and VR has been corrected in the text at various sections of the manuscript.

Abstract:

Should contain a bit of the research focus should be provided in the abstract. Just mentioning that “we brief about the development and progressive increase in applications of VR and AR technology in various areas of dental medicine and dental education” is not enough for readers to gain knowledge about the study. 

Author response: Thank you, corrected, abstract section, page number 01, line number 31 to 34, main document.

In addition, the authors should include a few sentences in the abstract that highlights the methodology, interesting findings, and implication of the study. If the authors are concerned about the length of the abstract after these adjustments, then, the leading sentences should be removed and taken to the introduction section.

Author response: Thank you, the correction is carried out in abstract section, page number 01, line number 34-37, main document

Methods and presentation:

I really like the presentation of the authors and enjoyed reading the different sections. However, I’m wondering about the style of the presentation. I was expecting to see a structure that showcased the research questions, methodology, results, discussion, and conclusion. While readers may take away some knowledge regarding the overview of AR and VR in dental medicine, it is not clear about the science of the study and what specific question is addressed. Unless these issues are addressed, the manuscript may look more like a piece of report than a scientific paper

Author response: Thank you, the correction is carried out. The manuscript is now representing the Introduction, Materials and Methods, Results, and Discussion sections.

Furthermore, this paper is a narrative review, hence a research question was not incorporated, the authors believe; that it is ideally a part of a systematic review.

Round 2

Reviewer 2 Report

This manuscript is designed as a Research paper with Results and Discussion, not a Review article which should have no experimental Results and there should be no Tables or Figures in the Discussion section. Thus, the entire manuscript needs to be restructure with the Results and Discussion section Titles (with Tables and Figures) should be replaced with appropriate Review-related topics (based on content). The Discussion section should be only text discussing the significance of the theory presented in the Review paper, not more information derived from reference sources. Also, Table 3 is still very low resolution and is in a Figure format. This should be made into a typed Table with text of normal size for tables. 

Author Response

Author response sheet

The corrections are carried out in relevant sections of the manuscript and are highlighted for clarity.

Reviewer 2.

  1. This manuscript is designed as a Research paper with Results and Discussion, not a Review article that should have no experimental Results and there should be no Tables or Figures in the Discussion section. Thus, the entire manuscript needs to be restructure with the Results and Discussion section Titles (with Tables and Figures) should be replaced with appropriate Review-related topics (based on content).

Author response: Thank you, corrected. The results and discussion section titles have been modified with review-related topics, pages 5-9, main document.

  1. The Discussion section should be only text discussing the significance of the theory presented in the Review paper, not more information derived from reference sources.

Author response: Thank you, the discussion section in the paper is replaced with the relevant text in the following area.

-Page 5, Paragraph 1, Lines 127-130.

-Page 8, Paragraph 1, Lines 251-252.

-Page 8, Paragraph 4, Lines 269-272.

-Page 8, Paragraph 6, Lines 283-284.

-Page 9-10, Paragraph 4, 327-332

  1. Also, Table 3 is still very low resolution and is in a Figure format. This should be made into a typed Table with the text of normal size for tables. 

Author response: Thank you, corrected. The table is omitted, and relevant text is added int the manuscript. Page number 5, line number 188, main document.

Reviewer 3 Report

The manuscript has improved. I suggest improving the figures (For instance: Fig.2.) and tables: Table 3.

Author Response

Author response sheet

The corrections are carried out in relevant sections of the manuscript and are highlighted for clarity.

Reviewer 3.

  1. The manuscript has improved. I suggest improving the figures (For instance: Fig.2.) and tables: Table 3.

Author response: Thank you, the correction is carried out, table 3 is omitted and text added on page number 5, line number 188, onwards. Figure 2 is now numbered “figure 1”  and amended to give conceptual insight in the manuscript, on page number 6.

Reviewer 4 Report

The authors have not addressed any of my comments. A review needs to offer conceptual insights (i.e., the “how”), not just merely listing others’ findings (i.e., the “what). The same problem exists in the figures as well as they are just lists of “what’, but offers no conceptual insight and does not need to be represented pictorially at all.

Author Response

Author response sheet

The corrections are carried out in relevant sections of the manuscript and are highlighted for clarity.

Reviewer 4:

The authors have not addressed any of my comments. A review needs to offer conceptual insights (i.e., the “how”), not just merely listing others’ findings (i.e., the “what). The same problem exists in the figures as well as they are just lists of “what’, but offers no conceptual insight and does not need to be represented pictorially at all.

Author response: Thank you.

  1. We appreciate the respected reviewer’s comment. The comment is really helpful and actually guided us in arranging the paper in order to be understood easily by the readers. The figures are removed from the manuscript as per the reviewer’s comment. Only 1 logical figure is present in the text, which has been modified extensively with conceptual insight. The text which was presented in the figures is added in relevant sections of the manuscript, line number 53-55, 85-90, 92-96, page number 2 and 3, main document. Moreover, Modified Figure 1, is present on page number 10.

  1. Thank you, for the correction regarding adding conceptual insight focusing on “how” and “why” AR/VR is helpful and when not handy, is carried out in the following areas of the text in the main manuscript.

-Page 4, Paragraph 2, Lines 121-123.

-Page 5, Paragraph 1, Lines 127-130.

-Page 8, Paragraph 1, Lines 251-252.

-Page 8, Paragraph 4, Lines 269-272.

-Page 8, Paragraph 6, Lines 283-284.

-Page 9-10, Paragraph 4, 327-332

Reviewer 5 Report

Overall, the authors have made some improvements in the manuscript but I think some improvement is still required. I have commented on these aspects which could make the manuscript publishable.

  1. In the abstract, it is not still clear what exactly the study focuses on. Why was this study conducted? what was the research gap? why should we read this paper? This information is more important in the earlier part of the abstract than the text in lines 26-34. In fact, I feel bored reading those leading sentences in that abstract which should be part of the introduction section instead.
  2. The author had described the methodology but it is still vague to the reader. I was just wondering, is the method a systematic literature review? If yes, then more details must be provided such that the study can be reproducible. For example, authors should include a table that shows how the search string was used in each of the databases utilized in this study, what was the search result for each database, etc.
  3. The authors could combine the "results" and "discussion" sections and improve the conclusion by adding the key implication of the study.
  4. Typos: I will recommend that the authors should proofread the manuscript thoroughly and correct English or typos. For example the ",," on line 161, ".." on line 405, references, etc.

Author Response

Author response sheet

The corrections are carried out in relevant sections of the manuscript and are highlighted for clarity.

Reviewer 5:

Overall, the authors have made some improvements in the manuscript, but I think some improvement is still required. I have commented on these aspects which could make the manuscript publishable.

  1. In the abstract, it is not still clear what exactly the study focuses on. Why was this study conducted? what was the research gap? why should we read this paper? This information is more important in the earlier part of the abstract than the text in lines 26-34. In fact, I feel bored reading those leading sentences in that abstract which should be part of the introduction section instead.

Author response: Thank you, corrected. Abstract section, Page 1, Lines 28-29, and 32-34, main document.

  1. The author had described the methodology, but it is still vague to the reader. I was just wondering, is the method a systematic literature review? If yes, then more details must be provided such that the study can be reproducible. For example, authors should include a table that shows how the search string was used in each of the databases utilized in this study, what was the search result for each database, etc.

Author response: Thank you. This is a review article, not a systemic review, therefore a precise methodology relevant to the paper is added as per reviewer comments.

  1. The authors could combine the "results" and "discussion" sections and improve the conclusion by adding the key implication of the study.

Author response: Thank you, corrected. The results and discussion sections have been combined. Conclusion improved Page 11, Lines 352-354.

  1. Typos: I will recommend that the authors should proofread the manuscript thoroughly and correct English or typos. For example the ",," on line 161, ".." on line 405, references, etc.

Author response: Thank you, the correction is carried out in various sections of the manuscript to improve English grammar and typo errors.

Round 3

Reviewer 2 Report

The authors have made significant improvements in the manuscript, but there are still some required revisions necessary to make it conform to journal requirements as follows.

There are a number of references that are out of sequence in the text that must be corrected. The following references in the Introduction are higher than reference numbers in Section 3 (which starts with reference 9):

L 111 - ref 28

L 114 - ref 29

L 117 - ref 30

L 123 - ref 28

Table 1 - column 1, place periods after et al. for all references followed by a space before the [ref#]

References without reference numbers following the citation:

L 206 Pulijala et al. [?] has no reference number

L 217 Won et al. [19?], remove [19] from the end of the sentence if this is the correct reference

L 241 Ran et al. [9?] move 9 from end of sentence to after reference citation

L 244 Osama et al [10?]

L 257 Jiang et al. [14?]

L 282 Llena et al. [18?]

L 288 Bruellmann et al. [25?]

Author Response

Point to point authors response to reviewer comments

Thank you for reviewing our manuscript. The corrections recommended by the respected reviewers are addressed in different sections of the manuscript. The track change is on in the manuscript. The corrections are further highlighted with distinct color for clarity. The detailed response to reviewer comments is described   low:

Reviewer 2.

The authors have made significant improvements in the manuscript, but there are still some required revisions necessary to make it conform to journal requirements as follows.

  1. There are a number of references that are out of sequence in the text that must be corrected. The following references in the Introduction are higher than reference numbers in Section 3 (which starts with reference 9):

L 111 - ref 28

L 114 - ref 29

L 117 - ref 30

L 123 - ref 28.

Author response: Thank you, corrected. The “Outcomes of Literature Search” section, pages 157-187, main document.

  1. Table 1 - column 1, place periods after et al. for all references followed by a space before the [ref#].

Author response: Thank you, corrected.  Table 1, Page number 5, main document.

  1. References without reference numbers following the citation:

L 206 Pulijala et al. [?] has no reference number

L 217 Won et al. [19?], remove [19] from the end of the sentence if this is the correct reference

L 241 Ran et al. [9?] move 9 from end of sentence to after reference citation

L 244 Osama et al [10?]

L 257 Jiang et al. [14?]

L 282 Llena et al. [18?]

L 288 Bruellmann et al. [25?]

Author response: Thank you, corrected. Page number 6-8, line number 218, 229, 234, 236, 269, 294, 300, main document.

Reviewer 3 Report

I suggest changing "dental surgery" or "dental surgeon". For example, in Table 1., Murugesan et al 2018 and Gujjar et al, 2019 have done the study related to dental surgery, I have read their paper and I know they wrote that in their papers, but, please, write the correct specialization because "dental surgery" is not a specialization. For example, periodontology, oral surgery, oral and maxillofacial surgery, oral medicine,... Please review this concern also in the text.

The paragraph between 316-330, only has one reference cited, number 6. This paragraph needs more cites along with the text.

I consider the manuscript has been improved a lot. Even though, I suggest to the authors, in order to have a review up-to-date, to include new papers published in 2022, as this one is related to one dental procedure during the COVID-19 pandemic:

Mladenovic R, AlQahtani S, Mladenovic K, Bukumiric Z, Zafar S. Effectiveness of technology-enhanced teaching methods of undergraduate dental skills for local anaesthesia administration during COVID-19 era: students' perception. BMC Oral Health. 2022 Feb 13;22(1):40. doi: 10.1186/s12903-022-02077-6. PMID: 35152899; PMCID: PMC8842892.

Author Response

Point to point authors response to reviewer comments

Thank you for reviewing our manuscript. The corrections recommended by the respected reviewers are addressed in different sections of the manuscript. The track change is on in the manuscript. The corrections are further highlighted with distinct color for clarity. The detailed response to reviewer comments is described  below:

Reviewer 3

  1. I suggest changing "dental surgery" or "dental surgeon". For example, in Table 1., Murugesan et al 2018 and Gujjar et al, 2019 have done the study related to dental surgery, I have read their paper and I know they wrote that in their papers, but, please, write the correct specialization because "dental surgery" is not a specialization. For example, periodontology, oral surgery, oral and maxillofacial surgery, oral medicine,... Please review this concern also in the text.

Author response: Thank you, corrected in table 1, page number 5, and in the manuscript text, main document.

Author response: Thank you. The author team appreciates the suggestion, the correction is carried out in the text, and references are added, page number 8,9, line number 314-342, main document.

  1. I consider the manuscript has been improved a lot. Even though, I suggest to the authors, in order to have a review up-to-date, to include new papers published in 2022, as this one is related to one dental procedure during the COVID-19 pandemic:

Mladenovic R, AlQahtani S, Mladenovic K, Bukumiric Z, Zafar S. Effectiveness of technology-enhanced teaching methods of undergraduate dental skills for local anaesthesia administration during COVID-19 era: students' perception. BMC Oral Health. 2022 Feb 13;22(1):40. doi: 10.1186/s12903-022-02077-6. PMID: 35152899; PMCID: PMC8842892.

Author response: Thank you, the correction is carried out, reference 31 cited in the manuscript.
